# R&B - Rhythm and Brain: Cross-subject decoding of musical tracks from human brain activity

Matteo Ferrante*
matteo.ferrante@uniroma2.it
University of Rome, Tor Vergata
Rome, IT

Matteo Ciferri*
matteo.ciferri@students.uniroma2.it
University of Rome, Tor Vergata
Rome, IT

Nicola Toschi
University of Rome, Tor Vergata
Rome, IT
Martinos Center For Biomedical
Imaging, MGH and Harvard Medical
Boston (MA), US

## ABSTRACT

Music is a universal phenomenon that profoundly influences human experiences across cultures. This study investigates whether musical tracks can be decoded from human brain activity measured with functional MRI (fMRI). Leveraging recent advancements in extensive datasets and pre-trained computational models, we constructed mappings between neural data and latent representations of musical stimuli. Our approach integrates functional and anatomical alignment techniques to facilitate cross-subject decoding, addressing the challenges posed by low temporal resolution and noise in fMRI data. We used the GTZan fMRI dataset, in which five participants listened to 540 musical tracks from 10 different genres while their brain activity was recorded. We used the CLAP (Contrastive Language-Audio Pretraining) model to extract latent representations of the musical tracks and developed voxel-wise encoding models to identify brain regions responsive to these stimuli. By applying a threshold to the correlation between predicted and actual brain activity, we identified specific regions of interest (ROIs) for music processing. Our decoding pipeline, primarily retrieval-based, employed ridge regression to map brain activity in the identified ROIs to the corresponding CLAP features. This enabled us to predict and retrieve the most similar musical tracks from the latent space based on neural data. The results demonstrated state-of-the-art identification accuracy, with our methods significantly outperforming existing approaches. The findings highlight the potential for neural-based music retrieval systems, opening new avenues for personalized music recommendations and therapeutic applications. Future work could explore the use of higher temporal resolution neuroimaging methods and more sophisticated generative models to further enhance the decoding accuracy and explore the neural underpinnings of music perception and emotion.

## KEYWORDS

Brain decoding, music, multimodal, neural music decoding, content retrieval

**ACM Reference Format:**
Matteo Ferrante*, Matteo Ciferri*, and Nicola Toschi. 2024. R&B - Rhythm and Brain: Cross-subject decoding of musical tracks from human brain activity. In . ACM, New York, NY, USA, 8 pages. https://doi.org/XXXXXXX.XXXXXXX

## 1 PROBLEM STATEMENT

Music universally permeates cultures around the globe, exerting a profound influence on the lives of all who can perceive its harmonies and rhythms. Its pervasive role across human societies is undeniable, yet the intricacies of how music impacts the human brain remain enigmatic. Music engages complex neurological pathways, triggering diverse emotional responses, evoking vivid episodic memories, and even interacting with various neurological disorders. These phenomena suggest that the relationship between music and brain function is both deep and multifaceted, warranting extensive scientific exploration [26]. In this paper, we investigate the intricate connection between brain activity and musical stimuli. Specifically, the research question we aim to address is whether (and to what extent) music tracks can be decoded from human brain activity measured with functional MRI while a subject is listening to these tracks,. The study of how the brain interprets and processes music has been a topic of classical inquiry within neuroscience [31]. However, recent advancements have revolutionized this field, allowing to use artificial intelligence (AI) to explore and decode brain patterns relative to a wide set of different kind of stimuli [28]. In this context, the emergence of extensive publicly available datasets coupled with robust, pre-trained computational models presents an unprecedented opportunity. These tools enable us to construct detailed mappings between neural data and latent, compact representations of external stimuli, such as images [6, 13, 14, 29, 32], videos [7], language [3, 11, 34], and notably, music [10]. These works propose several retrieval or generative pipelines to create a map between neural data and latent representations of external stimuli. The neural data is measured via functional magnetic resonance imaging (fMRI), magnetoencephalography (MEG), or electroencephalography (EEG), and the latent representations are obtained from pretrained models. The estimated latent space representations are further used for stimuli retrieval or conditioning of a generative model to generate images. Typically, these pipelines involve constructing mappings between these two spaces (brain data and latent representations of stimuli) and require subject-specific models, although multisubject brain representations or alignment and nonlinear mapping techniques have been presented [5, 15, 33].

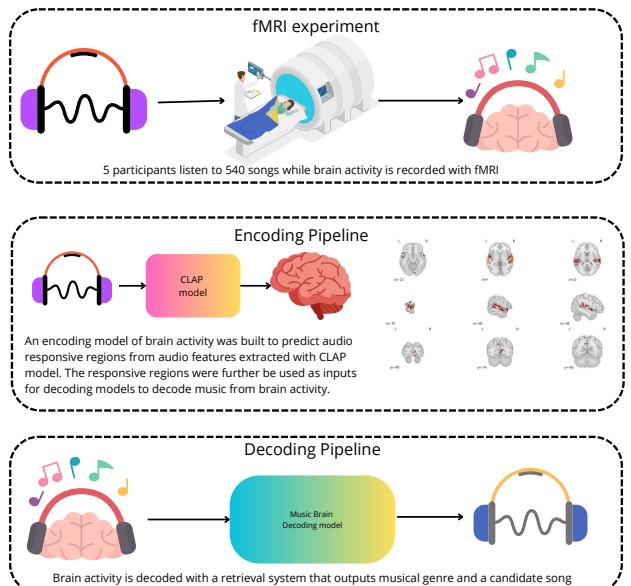

**Figure 1: Top Line: In the GTZan fMRI experiment, five participants listened to various musical tracks while their brain activity was monitored via fMRI, capturing neural responses to the music. Middle line: Our encoding pipeline starts with obtaining a latent representation of the music tracks using the CLAP model. We then develop voxel-wise encoding models to map the brain's response to these stimuli, identifying regions responsive to music based on the correlation between actual and predicted brain activities. Bottom line: Our decoding pipeline is retrieval-based. We train a model to predict CLAP features from brain activity data in the identified regions of interest (ROIs). Using these features, we search the CLAP latent space for the nearest musical tracks, selecting the closest five tracks as our retrieved samples.**

Mapping these complex relationships is both fascinating and informative, potentially offering insights into fundamental brain functions. Understanding the connection between music perception and neural responses could unlock novel avenues in diagnosing and treating neurological disorders. Moreover, it could enhance music therapy approaches, potentially leading to innovative treatments that harness the therapeutic properties of music [8, 19]. In this work, we aim to decode music from brain activity—translating the neural signals evoked by music perception into a high-fidelity representation of the musical stimulus that has elicited the former neural signals This objective challenges us to retrieve complex auditory information encoded within brain activity. The primary challenge lies in decoding a signal of inherently higher frequency from a subsampled neural signal, such as that obtained via fMRI, further complicated by the low pass filtering and time-shifting operated by the hemodynamic response function (HRF) [23]. Although the dataset depth of neural recordings is increasing and this opens up

new possibilities, the size is still relatively small typically comprising few subjects with anatomical and functional differences. To address these challenges, we first constructed encoding models to identify brain regions responsive to musical stimuli. We then aggregated brain activity across subjects using a functional alignment technique to facilitate a cross-subject decoding approach. This included aligning functional brain data and mapping the activity of the identified regions activity to the latent representations of music. These representations were derived using an open-source, multimodal pre-trained foundation model known as Contrastive Language-Audio Pretraining (CLAP), [12]. In the final part of our study, we compare representations of music reconstructed from brain activity with their real/original counterparts, employing a selection criterion that identifies the five closest matching representations as decoded candidates. The studies most closely related to our research include [4] and [10]. [4] demonstrates that representing musical stimuli through time-frequency decomposition and using linear and non-linear methods to reconstructing the same decomposition from brain activity is feasible and in this work they decode the auditory experience of specific songs using invasive intracranial encephalography (iEEG) data. This work exemplifies the potential of direct neural interfaces in music cognition research. Another pivotal study, [10], shares similarities with our approach in that it addresses the challenges of retrieval and generative music decoding using the same fMRI dataset used in this work. However, unlike our methodology, [10] uses subject-specific decoding pipelines based on anatomical atlases and proprietary models like MuLAN and MusicLM [2, 17]. In this paper, we advance the field by implementing a streamlined pipeline leveraging open-source models. Our approach begins by identifying brain regions that can be effectively modeled with latent representations of audio stimuli. Subsequently, we use brain activity from these regions to construct cross-subject decoding pipelines [15]. This strategy has surpassed previous methods on similar tasks, such as cross-subject image decoding, and has led us to achieve state-of-the-art results in decoding musical pieces. Figure 1 depicts a visual scheme of the whole pipeline. Through these methods, we aspire to refine our understanding of how music is processed within the brain and lay the groundwork for future explorations into the therapeutic potential of music in pathological settings.

## 2 MATERIAL AND METHODS

In this section, we describe the proposed method and the data we used. The data (Music Genre fMRI Dataset, curated by [27],) are publicly available and can be requested at https://openneuro.org/datasets/ds003720/versions/1.0.1. These data serves as a valuable resource for investigating the neural correlates of music perception and categorization in the human brain. The dataset comprises functional magnetic resonance imaging (fMRI) data collected from five subjects ("sub-001" to "sub-005") while they listened to music stimuli representing 10 distinct genres. The experimental protocol included 18 runs per subject, consisting of 12 training runs and 6 test runs. Each run is also associated with detailed information about each stimulus, including onset time, genre type, track name, and start and end times of excerpts from the original tracks. All stimuli have a duration of 15 seconds, including 2 seconds each

of fade-in and fade-out effects. The data are provided in intensity normalized form, i.e. after RMS normalization Our preprocessing pipeline comprises several key steps. To address potential artifacts in the fMRI data, we performed motion correction techniques. The motion correction process involves compensating for spatial displacements between successive volumes in the fMRI time series caused by subject motion. This correction is made at run level and is crucial for maintaining the spatial alignment of brain structures across time and ensuring the accuracy of subsequent neuroimaging analyses. We co-registered the fMRI data to the Montreal Neurological Institute (MNI) standard space using FSL's flirt and fnirt tools. Following co-registration (anatomical alignment), we applied detrending and standardization to the preprocessed fMRI data. Detrending eliminates low-frequency drifts from the data, which may result from scanner instabilities or physiological noise, while standardization normalizes the data to a zero mean and unit variance, ensuring uniformity and comparability across samples and subjects. Finally, we performed data averaging, aggregating fMRI data across multiple scans of repeated songs. Data averaging improves the signal-to-noise ratio and enhances the detection of consistent neural responses associated with the stimuli under investigation. We used FSL [18] for motion correction and co-registration and nilearn python library [1] to perform all other preprocessing steps. The final step is delaying the brain activity by 3 Repetition Times (TR) (4.5 s) in order to account for hemodynamic response and average the following 15 seconds of signal to obtain a neural representation for each track in our dataset. Our final dataset is composed of a total of 540 songs and processed fMRI pairs for each subject, divided into 480 for training and 60 for testing, as provided by data providers. In order to address the inherent variability in brain structure and function across different individuals, we explored three distinct methodologies for aggregating cross-subject data. The first method we implemented was **anatomical alignment**, which uses standard brain atlases to align brain imaging data from different subjects based on their anatomical landmarks. By mapping each subject's data to a common anatomical framework, we can directly compare and combine data across individuals, despite differences in brain size, shape, or orientation. This method is widely used in neuroimaging as it facilitates the direct comparison of localized brain activity across subjects. As described in the preprocessing section, we aligned all the images in the MNI neurological space using FSL. Moving beyond mere anatomical correspondence, our second method, **functional alignment**, aligns brain activity based on functional signals. This technique involves matching brain regions that demonstrate similar activity patterns during specific tasks or stimuli across different subjects. Unlike anatomical alignment, functional alignment accounts for individual variations in brain function topology that may not align with physical brain structures, making it particularly advantageous for studies where functional responses to complex stimuli, such as music, are the primary focus. To this end we leveraged the "hyperalignment" technique proposed by [16], which is based on Procrustes analysis. Lastly, given that in recent literature [5, 9, 15] linear layers are emerging as a useful tool to align neural representations in a common space, we employed ridge regression as a model to aggregate cross-subject brain data. This approach applies regularization to address multicollinearity in

high-dimensional datasets, which is typical of fMRI data. By introducing a penalty term, ridge regression shrinks the coefficients of less important variables, combining voxel-wise data from different subjects into a unified model, thus enhancing the stability and generalizability of our predictions. These techniques aim to enhance the robustness and accuracy of decoding models by aligning and integrating neural data from multiple subjects. Each method offers a unique approach to the challenge of intersubject variability, a common hurdle in neuroimaging studies.. Each of these methods was tested for its efficacy in improving the accuracy of our decoding models, with the goal of establishing a reliable approach to interpreting complex brain data in a multi-subject context. Our brain engages with music in intricate, non-linear ways, forming representations that support our cognitive processes.This complexity suggests that we need a model with a large representational capacity, which can be achieved through multimodality. A multimodal pre-trained model like CLAP (Contrastive Language-Audio Pretraining, [12]) may, therefore, mimic some aspects of how our brains process music. CLAP is a multimodal neural network trained with contrastive learning in the realm of audio and text processing. It is trained on a diverse set of audio and text pairs, learning to predict a shared vectorial representations between audio and text. The model employs a SWINTransformer [25] to extract audio features from log-Mel spectrograms and a RoBERTa model [24] to extract text ones, both projected into a shared latent space of identical dimensions. The similarity between audio and text features is measured using dot products, forming the basis for similarity scores. Using such a model, musical features can be extracted, leading to the transformation of audio stimuli into a vectorial representation. Fig A2 shows outputs of t-Distributed Stochastic Neighbor Embedding (t-SNE, [35]) to create a 2D representation of the music features (latent representation of music tracks obtained with CLAP) based on genre labels (Figure A2). The resulting t-SNE visualization provides insights into the distribution of music genres within the feature space, offering a qualitative understanding of how the CLAP model's representations align with genre labels. The first step of our study was to identify brain regions responsive to musical stimuli by constructing voxel-wise linear encoding models. These models map the latent representations of music onto voxel-wise brain activity. To assess the efficacy of each voxel's model, we used a cross-validation scheme, wherein the correlation between the predicted and actual brain activities of each voxel was measured. Model training incorporated a voxel-wise hyperparameter search for the regularization parameter $\alpha$. We explored a range of $\alpha$ values set on a logarithmic scale from $10^{-2}$ to $10^3$. After training, we use these encoding models to predict the whole brain activity, measuring the Pearson correlation between true and predicted activity. We established an empirical threshold for selection at a correlation of 0.1. This threshold was empirically chosen based during preliminary tests and was used to generate a brain mask. This mask delineates areas showing significant responsiveness to the musical stimuli because activity in these areas can be effectively predicted by using our musical features extracted from CLAP and our encoding models, thus providing a focused view of music-related brain activity. Following the identification of regions of interest (ROIs) responsive to music, our next objective was to construct a common model that could map the brain activity from these ROIs to the

latent representations of musical features. Under the assumption that selected regions of the brain can be modelled with CLAP, we could use the activity of this region to directly map from the activity itself to the musical features. This model aims to facilitate a translation process where the neural responses could potentially me directly mapped into musical features. Essentially, this step is about creating a predictive model where the brain's response could serve as a proxy for the music itself, illustrating a direct link between neural activity and musical perception. We therefore trained a Ridge regression between cross-subject brain activity in previously selected ROIs and CLAP features, again with hyperparameter optimization with the same grid. We then focused on optimizing the retrieval process within our testing dataset. For each estimated music track features, we selected the top-k closest elements based on the lowest L2 (Euclidean) distance between predicted and true musical features. This approach forms the basis of a straightforward retrieval pipeline, where the model searches for and retrieves the most similar musical pieces from the latent space, based on the neural activity recorded. This method demonstrates the potential to identify music directly from brain activity and offers a qualitative insight into the types of decoded music we can uncover. In our study, we measured the identification accuracy as described in the Brain2Music framework. Identification accuracy quantifies how accurately the predicted $d$-dimensional features correspond to the target features by computing the Pearson correlation coefficient between each pair of predicted and target features. In our case the features are the estimated and true CLAP features with dimensionality 512. The accuracy for each prediction is the proportion of correct identifications, where a correct identification occurs if the correlation for a given prediction is higher than for any other prediction. Identification accuracy is calculated as follows:

1. Construct a correlation matrix between the predicted embeddings and the target embeddings. Each element of this matrix, $C_{i,j}$, represents the Pearson correlation coefficient between the $i$-th predicted embedding and the $j$-th target embedding.

2. For each predicted embedding, check if the correlation with its corresponding target (diagonal element $C_{i,i}$) is greater than the correlations with all other targets (non-diagonal elements $C_{i,j}$ for $j \neq i$).

3. The identification accuracy for each prediction is calculated using an indicator function: $\text{id\_acc}_i = \frac{1}{n-1} \sum_{j=1}^{n} 1\left[C_{i,i} > C_{i,j}\right]$ where $1[\cdot]$ is the indicator function that returns 1 if the condition is true and 0 otherwise.

4. The overall identification accuracy is the average of individual accuracies across all predictions: $\text{id\_acc} = \frac{1}{n} \sum_{i=1}^{n} \text{id\_acc}_i$ Identification accuracy provides a quantitative measure of how well a model can distinguish between multiple classes or conditions directly from complex data like brain activity. Following [10], an identification accuracy of 90% implies that, on average, 10% of the predictions are incorrect. In a dataset with 60 examples, this means the correct music track is ranked sixth on average, with five other tracks mistakenly rated higher. For a more tangible demonstration of our results, qualitative examples of decoded music can be accessed at the provided URL https://shorturl.at/wcWkJ, where listeners can directly experience the output of our decoding process, offering an auditory validation of the model's performance.

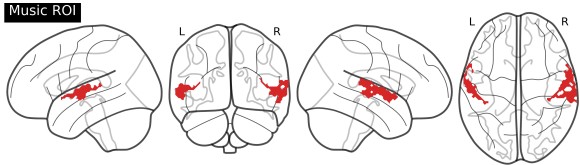

**Figure 2: Regions of interest corresponding to musically responsive areas were identified by applying a threshold to the correlations between predicted and actual brain activity. This process was part of a cross-validation procedure used in the encoding models.**

## 3 RESULTS

This study examined various embedding models and functional alignment strategies for classifying musical genres based on brain activity, highlighting significant improvements in accuracy and insights into music-responsive brain regions. The encoding models, tailored to detect regions responsive to musical stimuli, were successful in identifying a ROI composed by 833 voxels. Fig 2 shows the spatial position of the found ROI, which belongs to lateral and temporal regions of the brain. As shown in Table 1, our proposed methods with functional alignment techniques (denoted linear and hyperalign) demonstrated superior performance with identification accuracies of 0.9012 ± 0.01573 and 0.8805 ± 0.0231, respectively, outperforming other baselines and the anatomical alignment method. The linear alignment method, in particular, shows the highest performance, underscoring the efficacy of our linear modelling approach in this context.

**Table 1: Comparison of Test Identification Accuracy**

| Embedding | Test Identification Accuracy |
| --- | --- |
| SoundStream-avg | 0.674 ± 0.016 |
| w2v-BERT-avg | 0.837 ± 0.005 |
| MuLan$_{\text{text}}$ | 0.817 ± 0.014 |
| MuLan$_{\text{music}}$ | 0.876 ± 0.015 |
| Ours - anatomical | 0.7746 ± 0.01551 |
| **Ours - hyperalign** | **0.8805 ± 0.0231** |
| **Ours - linear** | **0.9012 ± 0.01573** |

The confusion matrix shown in Fig 3 illustrates the model's capability to classify musical genres based on brain activity, with a notable concentration of correct predictions along the diagonal. Classical and jazz genres showed high accuracy with minimal confusion, suggesting that they correspond to distinct neural representations. However, genres like metal and disco exhibited more confusion, likely due to overlapping music features that are less distinguishable by the model. For example, the confusion between disco and metal may arise from similar rhythmic patterns or instrumentation that blur genre-specific boundaries in neural encoding. Figure A3 shows the similarity between the retrieved musical tracks and the original genre stimulus. Within the retrieved cluster, the exact stimulus is found very often, emphasizing the effectiveness of

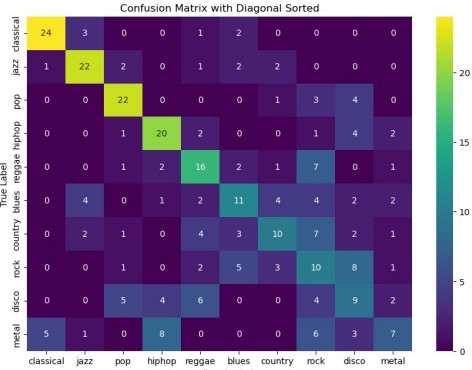

**Figure 3: This confusion matrix shows our model's accuracy in classifying musical genres based on fMRI data from five participants. Diagonal elements represent correct predictions for each genre, while off-diagonal elements indicate misclassifications. In the experiment setup for testing runs, each genre has 30 tracks, evenly distributed across the subjects; a number 30 in the main diagonal represents 100% accuracy. The model performs well for classical, jazz, and pop genres, with minimal confusion, while disco and metal show higher misclassification rates, likely due to overlapping music features. The matrix highlights the effectiveness of the cross-subject decoding pipeline and areas for improvement.**

the pipeline. Given feature overlap, it is common to encounter different genres in the retrieved group of music tracks compared to the stimulus, although always within genres that exhibit shared acoustic patterns. The functional alignment techniques, significantly enhanced the identification accuracy respect other baselines. This improvement indicates that aligning functional brain data across subjects, while preserving individual differences in brain anatomy, allows for more accurate generalizations when decoding music genres from brain activity. The technique effectively harnesses shared information across different subjects, thereby boosting the overall model's performance. Compared to existing studies, such as those using basic MuLan or SoundStream embeddings [10, 17], our methods provide a clear advantage in music track retrieval and genre classification accuracy. Previous studies often did not account for individual variations in brain anatomy and function as effectively, which our hyperalign and linear methods address directly. The results from this study not only reinforce the utility of advanced machine learning techniques in neuroscience but also pave the way for more personalized and accurate interpretations of brain activity in response to complex stimuli like music. Future work could explore deeper neural network architectures or alternative machine learning models that might further refine the accuracy of musical genre classification from brain imaging data.

## 4 SIGNIFICANCE

The findings of this study provide compelling evidence that decoding music from cross-subject neural activity is not only feasible but also remarkably accurate.. This opens up numerous possibilities for understanding the cognitive processing of music and its

applications, ranging from therapeutic practices to advanced brain-computer interfaces. The successful decoding of music genres from brain activity suggests profound implications for cognitive neuroscience and psychological studies. By associating specific genres with distinct patterns of brain activation, researchers can further explore how these patterns correlate with cognitive functions, emotional states, and individual preferences. This understanding could eventually lead to personalized music interventions designed to manage various psychological conditions such as anxiety, depression, and stress. Further refinement of this process could lead to neural-guided recommendation systems, allowing individuals to receive personalized music suggestions based on neural similarities with tracks they enjoy or those that evoke specific emotions. Our analysis achieved results in line with [27], and this further shows that certain genres, like classical and jazz, are more distinctly encoded in the brain, possibly due to their unique structural and rhythmic complexities which might engage specific neural pathways. However, the confusion between closely related genres like rock and metal highlights the challenges in distinguishing between similar auditory stimuli and suggests a need for more refined modelling techniques that can capture subtle nuances in music perception. This research has potential applications in music therapy. Understanding the neural basis of the influence of music on emotion and cognition can improve therapeutic protocols, as noted by [30, 31]. Precise genre-specific neural decoding could tailor therapies to individual needs. Further research into music and emotions, as explored by [20–22], could decode emotional content from brain activity, aiding in the development of a neural recommendation system for personalized music suggestions based on emotional and neural states. Looking forward closer in time, the decoding techniques used in this study could be extended to generative music systems, potentially leading to innovative applications in creating music from brain activity, including musical imagery. At the time of writing, the primary reason we are focusing on retrieval rather than generation is due to the limitations posed by the low temporal resolution of fMRI acquisition. This limitation constrains the possibility of generating music based on neural dynamics, which may be achievable with other neural activity measures like iEEG or MEG. However, a particularly intriguing prospect is to replace the retrieval module with a generative one, especially by combining music decoding with imagery. Imagine an artist entering the scanner and envisioning a music track to be decoded through this process. The resulting piece could be seen as a collaborative creation between the artist's imagination and artificial intelligence, potentially giving rise to a new art form where learned musical priors are transformed and utilized by neural decoding models to produce unique artistic expressions. Such systems would not only deepen our understanding of the creative processes that underpin music generation but also open the door to innovative forms of artistic expression that are directly influenced by neural dynamics. Despite these advancements, several limitations remain. The neural signals used in this study are inherently noisy and are only a subsampled representation of brain activity, which limits the detail and accuracy of the music that can be reconstructed. Rhythmic elements, particularly those at fine temporal resolutions, remain challenging to decode accurately due to the limitations in the temporal resolution of fMRI technology. Moreover, the extensive scanning time

required for collecting sufficient data is a practical limitation that could restrict the use of these techniques in everyday applications. Future research could explore the use of alternative neuroimaging methods, such as electroencephalography (EEG) or intracranial EEG (iEEG), which offer higher temporal resolution and could potentially provide more detailed insights into the neural encoding of music. Additionally, the development of more sophisticated generative models that can better handle the complexity and variability of neural data represents a promising direction for both academic research and practical applications in neuromusicology.

## 5 ACKNOWLEDGEMENTS

This work was supported by NEXTGENERATIONEU (NGEU) and funded by the Italian Ministry of University and Research (MUR), National Recovery and Resilience Plan (NRRP), project MNESYS (PE0000006) (to NT)– A Multiscale integrated approach to the study of the nervous system in health and disease (DN. 1553 11.10.2022); by the MUR-PNRR M4C2I1.3 PE6 project PE00000019 Heal Italia (to NT); by the NATIONAL CENTRE FOR HPC, BIG DATA AND QUANTUM COMPUTING, within the spoke "Multiscale Modeling and Engineering Applications" (to NT); the EXPERIENCE project (European Union's Horizon 2020 Research and Innovation Programme under grant agreement No. 101017727); the CROSSBRAIN project (European Union's European Innovation Council under grant agreement No. 101070908).

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

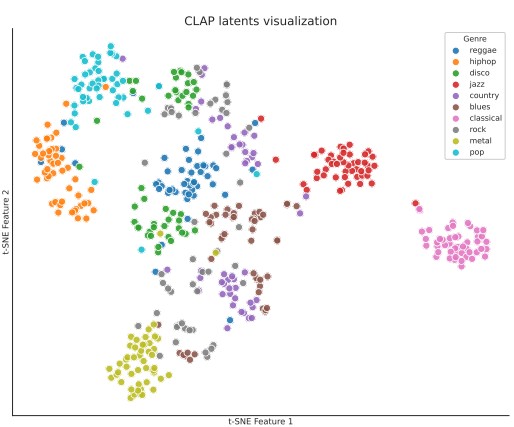

**Figure A2: Two-dimensional t-SNE representation of CLAP latents of musical tracks, coloured by different musical genres.**

# A APPENDIX

## A.1 Decoding in time

In the main experiment, we used the average the 10 samples acquired for every voxel during the listening of a track in order to obtain a neural signature of a given track. Another possible interesting research question is when, after the stimulus onset, in the brain we could observe a peak in performances for music decoding. To ask this question we evaluated the neural responses elicited by each time sample (each TR). This analysis is exactly the same as the one described in the main paper, except that instead of using the averaged brain activity over 15s as input for the decoding model we're using the instant brain activity at each time point. Subsequently, we built a decoding-in-time representation (Figure A1) showing the sample exhibiting the highest degree of engagement in processing musical stimuli. This approach not only unveils the specific temporal dynamics underlying music perception within the brain but also sheds light on the samples most prominently involved in this process.

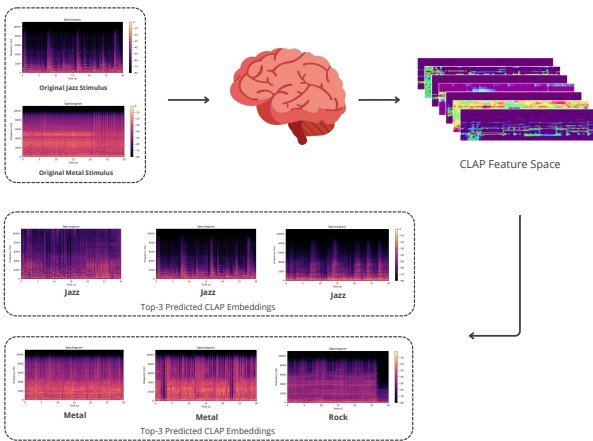

**Figure A3: This figure showcases the spectrograms of original musical stimuli (jazz and metal) and the top-3 predicted CLAP embeddings obtained from the Ridge regression decoding model. The left side displays the spectrogram of the original jazz stimulus, while the right side shows the spectrogram of the original metal stimulus. Below each original stimulus, the top-3 predicted embeddings are illustrated. For the jazz stimulus, the predicted embeddings were all identified as jazz. For the metal stimulus, the top-3 predictions included two metal and one rock embedding. This comparison highlights the model's ability to accurately predict musical genres from brain activity, while also illustrating occasional genre misclassification, particularly in more complex or overlapping genre spaces.**

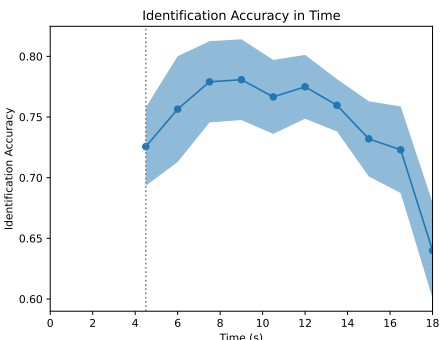

**Figure A1: This figure depicts the identification accuracy of the music decoding model over a time course of 18 seconds. The y-axis represents the identification accuracy, while the x-axis represents the time in seconds. The graph shows a trend of increasing identification accuracy as time progresses, reaching a peak towards the later part of the time window. This indicates that the model's ability to accurately decode musical genres from brain activity improves with longer exposure to the musical stimuli, suggesting that prolonged neural engagement with the music enhances the decoding performance.**