# OpenReview forum: "R&B -  Rhythm and Brain: Cross-subject decoding of musical tracks from human brain activity"
_KDD.org/2024/Workshop/AIDSH — KDD-AIDSH 2024 Oral_

### Official Review · Reviewer_T5Tt · 2024-06-14
**Decoding Musical Tracks from fMRI Brain Activity Using Cross-Subject Alignment Techniques**

**Rating:** 8
**Confidence:** 3

**Review:**

This manuscript explores how to decode musical tracks from brain activity measured by functional magnetic resonance imaging (fMRI). The research combines functional and anatomical alignment techniques, using the CLAP model to extract musical features and constructing voxel-wise encoding models to identify brain regions responsive to music. Through these methods, the study achieves state-of-the-art results in cross-subject music decoding, demonstrating the potential of neural-based music retrieval systems and paving the way for personalized music recommendations and therapeutic applications.

Overall, the issues and methods proposed in this manuscript are interesting. I agree to accept this manuscript, but please break the paragraphs in the original text. The current state causes some difficulty in reading. As it stands, the small number of subjects in the dataset may limit the persuasiveness of the results. If possible, I hope the authors will validate the proposed methods with a larger dataset in the future.

---

### Official Review · Reviewer_nGos · 2024-06-16
**Amazing study about brain and music**

**Rating:** 9
**Confidence:** 4

**Review:**

Summary:The paper investigates the possibility of decoding musical tracks from functional MRI (fMRI) brain activity data. The authors leverage advancements in multimodal datasets and pre-trained models to map brain activity to latent representations of musical stimuli. This study extends the potential for neural-based music retrieval systems and suggests applications in personalized music recommendations and therapeutic interventions.

Strengths: The paper introduces a novel methodology for decoding music from brain activity, utilizing a cross-subject framework that improves generalizability across different individuals.

In addition, the authors provide comprehensive analyses, including voxel-wise encoding models and rigorous statistical validation, which strengthen the conclusions drawn from the study. Besides, as a R&B lover, I love this title and paper. it is really amazing.

Weakness: While the paper mentions future work, it does not extensively discuss the current limitations of the methods, such as the low temporal resolution of fMRI and the challenges in handling individual variability in brain anatomy and function.

The approaches used might not scale well to larger, more diverse populations given that the study was conducted with only five participants, which could affect the external validity of the findings.

---

### Decision · Program_Chairs · 2024-06-28

Accept (Oral)